# Effects of FODMAPs and Gluten on Gut Microbiota and Their Association with the Metabolome in Irritable Bowel Syndrome: A Double-Blind, Randomized, Cross-Over Intervention Study

**DOI:** 10.3390/nu15133045

**Published:** 2023-07-05

**Authors:** Elise Nordin, Per M. Hellström, Johan Dicksved, Erik Pelve, Rikard Landberg, Carl Brunius

**Affiliations:** 1Department of Life Sciences, Food and Nutrition Science, Chalmers University of Technology, SE-41296 Gothenburg, Sweden; rikard.landberg@chalmers.se (R.L.); carl.brunius@chalmers.se (C.B.); 2Department of Medical Sciences, Gastroenterology/Hepatology, Uppsala University, SE-75185 Uppsala, Sweden; per.hellstrom@medsci.uu.se; 3Department of Animal Nutrition and Management, Swedish University of Agricultural Sciences, SE-75007 Uppsala, Sweden; johan.dicksved@slu.se; 4Department of Anatomy, Physiology and Biochemistry, Swedish University of Agricultural Sciences, SE-75007 Uppsala, Sweden; erik.pelve@slu.se

**Keywords:** irritable bowel syndrome, FODMAPs, gluten, gut microbiota, SCFAs, metabolome, multiomics

## Abstract

Background: A mechanistic understanding of the effects of dietary treatment in irritable bowel syndrome (IBS) is lacking. Our aim was therefore to investigate how fermentable oligo- di-, monosaccharides, and polyols (FODMAPs) and gluten affected gut microbiota and circulating metabolite profiles, as well as to investigate potential links between gut microbiota, metabolites, and IBS symptoms. Methods: We used data from a double-blind, randomized, crossover study with week-long provocations of FODMAPs, gluten, and placebo in participants with IBS. To study the effects of the provocations on fecal microbiota, fecal and plasma short-chain fatty acids, the untargeted plasma metabolome, and IBS symptoms, we used Random Forest, linear mixed model and Spearman correlation analysis. Results: FODMAPs increased fecal saccharolytic bacteria, plasma phenolic-derived metabolites, 3-indolepropionate, and decreased isobutyrate and bile acids. Gluten decreased fecal isovalerate and altered carnitine derivatives, CoA, and fatty acids in plasma. For FODMAPs, modest correlations were observed between microbiota and phenolic-derived metabolites and 3-indolepropionate, previously associated with improved metabolic health, and reduced inflammation. Correlations between molecular data and IBS symptoms were weak. Conclusions: FODMAPs, but not gluten, altered microbiota composition and correlated with phenolic-derived metabolites and 3-indolepropionate, with only weak associations with IBS symptoms. Thus, the minor effect of FODMAPs on IBS symptoms must be weighed against the effect on microbiota and metabolites related to positive health factors.

## 1. Introduction

Irritable bowel syndrome (IBS) is a common condition affecting around 3–5 percent of the population in Western countries and is defined according to the Rome IV criteria by recurring abdominal pain in relation to stool irregularities [1]. The mechanisms behind IBS are poorly understood, but changes in gut microbiota composition, intestinal barrier function, enteroendocrine cell population, low-grade inflammation and gut–brain axis modulations are believed to play a role [2,3,4,5]. Diet has in recent years been associated with symptom management in IBS and the so-called low-FODMAP diet (fermentable oligo-, di-, monosaccharides, and polyols) has become popular, due to its reported ability to reduce IBS symptoms [6]. The low-FODMAP diet is assumed to decrease microbial fermentation with less gas production and fewer osmotic metabolites, leading to reduced symptoms, predominantly abdominal distension, and pain [7]. FODMAPs consist of fermentable carbohydrates, which constitute an energy source for bacteria and can be fermented into metabolites such as short-chain fatty acids (SCFAs). Among these, acetate, propionate, and butyrate are derived from colonic fermentation of dietary fiber [8] whereas isobutyrate and isovalerate are derived from fermentation of proteins [9]. Removal or addition of FODMAPs may elicit alterations in the gut microbiota composition and/or activity and subsequent metabolite formation [10]. FODMAPs have consistently been found to increase *Bifidobacterium* [11,12,13,14,15], but such effects have not been related to the severity of IBS symptoms [11,12,14]. Other effects of FODMAPs on the gut microbiota have been inconclusive [16]. The impact of FODMAPs on fecal SCFAs has been inconsistent [12,14,15,17,18] and their effect on plasma concentrations has previously not been reported.

A gluten-free diet has also been advocated for alleviation of IBS symptoms, but results from clinical trials are inconsistent [19,20,21,22] and potential mechanisms unclear [23]. Studies examining a gluten-free diet in both diseased and healthy humans have shown alterations in the gut microbiota composition [24,25,26], mainly with reduced levels of *Bifidobacterium* [25,26]. However, these results are believed to relate to the simultaneous reduction of dietary fiber and not the intake of gluten per se [26,27]. Interestingly, a study on healthy subjects consuming a diet rich in gluten found no effects on the gut microbiota composition [26]. However, no such study has been performed in people with IBS. In addition, conflicting results have been reported for SCFAs in relation to gluten intake [26,28].

Combining omics measurements at multiple levels of biology, i.e., multiomics, has emerged as an approach for studying complex biological processes [29,30] and has the potential to discover novel disease mechanisms [31]. It may also offer improved precision in the prediction of disease conditions and metabolic perturbations. To gain mechanistic insight into effects related to gut microbiota composition, functional analysis is required, e.g., by genomic and/or metabolomic analysis [29,32]. It is well known that a large portion of the metabolome is connected to the microbiome [33,34,35,36]. However, much is still unknown and more efforts are needed in identifying the relationships between microbes and metabolites [26,34,37]. Only a few studies have investigated the combined effects of the gut microbiota and metabolites in IBS and FODMAPs [11,12]: one study reported that *Porphyromonadaceae* spp. correlated strongly with urinary histamine [11], while another study found no meaningful correlations among bacterial genera, fecal SCFAs, cytokines and IBS symptoms [12]. To our knowledge, no such study has been reported for IBS and gluten.

We have previously performed a double-blind, placebo-controlled, randomized three-way crossover trial with week-long interventions with FODMAPs, gluten or placebo in people with IBS by estimating symptoms with the IBS severity scoring system (IBS-SSS) [22]. The FODMAP intervention resulted in a modest increase in IBS symptoms compared to the placebo, whereas there was no observable difference between the gluten and placebo interventions at group level [22]. Furthermore, the FODMAP intervention affected mainly microbial-derived metabolites in plasma such as decreased levels of bile acids, an increase in phenolic-derived metabolites and 3-indolepropionate (IPA), and both down- and upregulation of unknown phenyl sulphates [38]. Of note is IPA, a gut microbiota-derived metabolite which in previous trials has correlated with dietary fiber intake [39,40]. In our trial, IPA correlated with the IBS symptom of abdominal pain and quality of life, albeit weakly [38]. Mechanisms may involve microbiota shifting the metabolism of tryptophan towards the kynurenine pathway [41,42], or it may be that IPA itself affects the kynuerine pathway [43], which is suggested to be involved in the pathogenesis of IBS [44]. Gluten was shown to affect lipid-related metabolites only modestly, including carnitine derivates, an acyl-CoA derivate, a medium chain fatty acid, and an unknown lipid, but with no meaningful correlation with IBS symptoms.

Since there are inconsistencies in the effects of FODMAPs and gluten on microbiota and SCFAs, and there is also a general lack of multiomics analysis in IBS and dietary trials, the aim of the present exploratory work was to investigate the effects of FODMAPs, gluten or placebo on both gut microbiota composition, fecal and plasma SCFAs and their relationship to IBS symptoms and the metabolome. We hypothesized that the FODMAP challenge would affect gut microbiota composition and that specific genera would be correlated with metabolites derived from dietary fiber fermentation and plant-based foods, including SCFAs, bile acids, phenolics and indole compounds. Furthermore, we hypothesized that such alterations might be related to IBS symptoms, as weakly indicated in previous studies [11,12,16,45,46]. Given the weak corresponding associations for gluten in our previous study [22] and in the literature [20,23], we hypothesized that the corresponding associations for gluten would be weak, although highly relevant for evaluation, since this topic has scarcely been studied.

## 2. Materials and Methods

### 2.1. Participants and Study Design

The double-blind, placebo-controlled three-way crossover study was conducted in September 2018–June 2019 in Uppsala, Sweden and has been described in detail elsewhere [22]. The main eligibility criteria of the study were: moderate-to-severe IBS (participants were diagnosed according to the Rome IV criteria and subtyped into constipation, diarrhea or mixed IBS [1]), no other gastrointestinal disease, BMI between 18.5 and 38 kg/m^2^ and age 18–70 years. In brief, 110 participants with IBS were included, 103 of which completed the trial (90 females, 13 males, 46 ± 15 years of age, and BMI 24 ± 4 kg/m^2^; further details are described in Table 1 [22]). Both the study personnel and participants were blinded. During the seven weeks of the study, participants consumed a so-called low-impact diet, i.e., excluding gluten and having minimal consumption of FODMAPs, guided by a dietitian. Between the two initial run-in weeks on a low-impact diet, a single combined challenge test with FODMAPs and gluten was carried out, following which blood samples were drawn at regular intervals during four hours, for later analysis. Thereafter, participants underwent one-week interventions with FODMAPs, gluten and placebo, respectively, with one washout week in between. At the end of each week 2–7, participants returned the questionnaire, along with a feces sample, anthropometric measures were registered, and fasting blood samples were drawn, reflecting exposures during each previous week. A visualization of the study design is presented in Appendix A. Participants were randomized into the sequences CBA, ACB, BAC in blocks of 12 (A = FODMAPs, B = gluten, C = placebo). Randomization was performed by personnel not involved in the study. Allocation was delivered to the study site 1–3 days before the participant started the study. Trial registration: www.ClinicalTrials.gov (NCT03653689) 31 August 2018, accessed on 1 May 2023. The manuscript was prepared in agreement with CONSORT guidelines (https://www.consort-statement.org, accessed on 1 May 2023).

### 2.2. Dietary Interventions

The dietary interventions have been described in detail elsewhere [22]. In brief, the daily dose of FODMAPs during the FODMAP intervention was 50 g (lactose 15.7 g, fructose 19.5 g, fructo-oligosaccharides 7 g, galacto-oligosaccharides 1.5 g, sorbitol 5.4 g, and mannitol 1.8 g) and the corresponding dose for the gluten intervention was 17.3 g. The doses corresponded to 150% of the daily intake for someone in the Australian population [47], except for lactose and gluten, which were based on 150% of the intake for someone in the Swedish population [48]. To mimic the sweetness of the FODMAP intervention, the placebo intervention consisted of 18 g sucrose. Due to a similar reason, 24 g icing sugar was added to the gluten intervention. The interventions were delivered in powder form added to rice porridge, and the daily dose was divided into three servings distributed during the day. Self-reported consumption of >80% per week was considered compliant. The daily intake and nutrient content of the intervention foods are presented in Appendix A.

### 2.3. Questionnaires

At each visit to the clinic the following was registered: adherence to overnight fasting routines and compliance with avoidance of vigorous physical activity and alcohol consumption during the preceding 24 h. Questionnaires were handed in at each visit to the clinic and included the validated IBS severity scoring system (IBS-SSS) [49], short form 36 version 2 [50] reflecting health and quality of life, and a bowel emptying diary. The IBS-SSS consists of five items: severity of abdominal pain, frequency of abdominal pain, abdominal distension, dissatisfaction with bowel habits and interference with quality of life. Each item is rated from 0–100 on a visual analog scale (VAS) and the total IBS-SSS score ranges from 0–500. The health and quality-of-life questionnaire measures the categories physical functioning, role—physical, bodily pain, general health, vitality, social functioning, role—emotional, mental health, mental component score and physical component score. The bowel emptying diary collects information on stool consistency (measured using the Bristol Stool Form Scale (BSFS): BSFS1-2 = hard, BSFS3-5 = normal, BSFS6-7 = loose [51]), frequency, pain in relation to bowel emptying (VAS 0–100), whether medication was needed, and whether bowel emptying referred to spontaneous bowel movements (SBMs) or completely spontaneous (SCBMs) [52].

### 2.4. Collection of Fecal Samples

Participants collected feces samples as close to their upcoming visit as possible and stored them in their home freezer until visiting the clinic. During transport to the clinic, samples were kept in a box with freezing blocks. At the clinic, samples were frozen at −20 °C. Within 7 days of arrival, samples were transferred to a −80 °C freezer.

### 2.5. Gut Microbiota Data

In total, 621 feces samples were analyzed. Three-level constrained randomization assured that the participant order was randomized, and that samples from the same participant ID were analyzed next to each other in the same batch, that the order of the interventions for the participant ID were randomized, and that the order of intervention and preceding run-in/washout week was also randomized. DNA was isolated from the feces samples with QIAamp DNA Stool Mini kit (Qiagen, Hilden, Germany) with an additional bead-beating step to increase efficiency in the lysis of bacterial cell walls. The bead beating was carried out with 0.1-mm Silica/Zirconia beads for 2 × 45 s using a Precellys homogenizer (Bertin Technologies, Montigny-le-Bretonneux, France). Purified DNA was stored at −20 °C until analysis. The 16S RNA gene amplicon libraries were generated by amplifying the V3-V4 region of 16S rRNA genes with the primers (341F 5′-CCTACGGGAGGCAGCAG-3′ and 806R 5′-GGACTACNNGGGTATCTAAT-3′). PCR results were purified with GeneJET Gel Extraction Kit (Thermo Fisher Scientific, Waltham, MA, USA). Sequencing libraries were generated using NEB Next^®^ Ultra™ DNA Library Prep Kit for Illumina (New England Biolabs, Ipswich, MA, USA) and library quality was assessed on the Qubit@ 2.0 Flourometer (Thermo Fisher Scientific) and Agilent Bioanalyzer 2100 system (Agilent, Santa Clara, CA, USA). The amplicon library was sequenced on an Illumina HiSeq 2500 platform, generating 250 bp paired-end reads (Illumina, San Diego, CA, USA).

The raw demultiplexed reads from the sequencing were processed using DADA2 to denoise (filterAndTrim parameters: maxN = 0, maxEE = c (2,2), truncQ = 2, rm.phix = TRUE, compress = TRUE), dereplicate reads, merge paired-end reads and remove chimeras [53]. The table of amplicon sequence variants (ASVs) were assigned to reference sequences against the SILVA rRNA database with the naive Bayesian classifier called with the assign Taxonomy command [54,55], release 138, formatted for dada2 by B. Callahan (https://benjjneb.github.io/dada2/training.html, accessed on 1 May 2023). The Phyloseq R package was used to construct ASV frequency tables [56]. The ASVs were filtered on the condition of being present in ≥10 sequences and in ≥5 samples. Alpha diversity measures of richness, Shannon’s diversity index, Simpson’s index, inverse Simpson’s index, and visualization of beta diversity with principal coordinates analysis (PCoA) were calculated using the ASV data.

### 2.6. Analysis of Short-Chain Fatty Acids

In total, seven SCFAs, succinate, and the medium-chain fatty acid caproate were purchased; formate (Scharlau), acetate (Honeywell), sodium propionate, isobutyrate, and valerate (Alfa Aesar), sodium butyrate, isovalerate and caproate (Sigma Aldrich, Saint Louis, MA, USA), and sodium succinate (Acros). Derivatizing reagents 3-nitrophenylhydrazine hydrochloride (3-NPH), N- (3-dimethylaminopropyl)-N’-ethylcarbidimide hydrochloride (EDC-6), D- (-) quinic acid, HPLC grade pyridine, LiChrosolv methanol hypergrade for LC-MS, and LiChrosolv LC-MS grade water were purchased from Sigma Aldrich, and LC-MS grade acetonitrile from Fisher Scientific. 13C6-3NPH-HCl was synthesized by IsoSciences Inc. (King of Prussia, PA, USA) (catalogue 13309).

Stock solutions of 4 mM acetic and formate, 2 mM propionate, and 1 mM for the remaining SCFA were prepared in 75% MeOH. Thereafter, a standard mix with 1:10 dilution (10% MeOH) of the SCFA was made. Formate and acetate standards were prepared in the concentration 3.2 µM-0.63 nM, propionate 3.2 µM-0.31 nM and the remaining SCFAs in the range 0.8 µM- 0.16 nM. In total, five blanks (LiChrosolv) and three quality control (QC) samples in triplicates were included in each batch. Two QC samples were long-term pooled plasma samples, the third was pooled plasma of samples from the current dataset. The reagents 3-NPH 200 mM, quinic acid 200 mM, and 120 mM EDC-6 in 6% pyridine were prepared in 75% MeOH. The 13C-labelled internal standard for all SCFAs was prepared by weighing 1 g of 13C6-3NPH HCl, adding 50 μL of the 1:10 standard mix solution, 25 μL of 120 mM EDC-6 and 25 μL 75% MeOH. The dilution was mixed and shaken in a cold room for 4 h. When finished, 25 μL quinic acid was added, and the sample was shaken at 1600 rpm (multitube vortexer DUX-2500) in RT for 45 min. Finally, it was diluted to 100 mL with 10% MeOH.

#### 2.6.1. Sample Preparation of Plasma Samples

In the first step, 10 μL of either sample plasma, QCs, water for blanks, or standards, together with 10 μL 3-NPH and 10 μL EDC-6, and 60 μL 75% MeOH were added to a tube and incubated at room temperature (RT) for 45 min with shaking (1600 rpm). Thereafter, the reaction was quenched by the addition of 10 μL quinic acid, followed by 15 min of incubation in RT with shaking, and centrifugation at 16,000× *g* 4 °C for 5 min. Supernatant (75 μL) was transferred to a new tube and 925 μL 10% MeOH was added. Samples were vortexed and centrifuged at 16,000× *g* 4 °C for 5 min. In the final step, 100 μL of sample and 100 μL of 13C-labeled internal standard solution were transferred to an LC glass vial.

#### 2.6.2. Sample Preparation of Fecal Samples

For fecal samples, 20 mg sample (dry weight) together with 1 mL LiChrosolv Water was added to a tube and shaken for 15 min at 1600 rpm, followed by centrifugation at 16,000× *g* 4 °C for 5 min. Thereafter, 10 μL of the fecal water was prepared according to the same protocol as the plasma samples until the final step, when the samples were added to the vials. At this step, 25 μL of sample (instead of 100 μL as with the plasma) and 75 μL of 10% MeOH were added, and finally, 100 μL of 13C-labeled internal standard was transferred to the vial. The concentration of formate was below the limit of quantification and was therefore not included in the data analysis.

#### 2.6.3. Parameters of LC-MS

Samples were analyzed on an ExionLC instrument (AB Sciex, Stockholm, Sweden) with a 6500+ QTRAP triple-quadrupole mass spectrometer (AB Sciex, Stockholm, Sweden), equipped with an APCI ion source, operated in the negative-ion mode. Analytes were separated on a Waters Acquity UPLC BEH C18 column (1.7 um, 2.1 mm × 150 mm) (Birketoften 13, 3500 Værløse, Denmark) using a mobile phase consisting of MilliQ water (filtered through an LC-Pak Polisher (Merck, Molsheim, France) (A) and acetonitrile (B) delivered in a gradient: 0.5% solvent B 3 min, at 3 min, 2.5% solvent B ramping linearly to 17% solvent B at 6 min, then to 45% B at 10 min, and 55% B at 13 min, at a flow rate of 400 μL/minute. The column oven was set at 40 °C, the curtain gas and the ion source 50 psi, and the collision gas was set to medium. The ion source was 500 °C, nebulizer current −3 uA and the injection volume 12 μL. The autosampler was kept at 4 °C. Samples from the same individual were run in the same batch. After every five injections, a solvent blank was run (75% MeOH) and after every 15 injections, two solvent blanks were run. Multiple-reaction monitoring (MRM) was used to detect analytes. Two MRM transitions were used (Q1/Q3) per analyte, one serving as a method of quantitation and the other acting as a qualifier. Data were acquired using the Analyst 1.7 software and were processed with the MultiQuant 3.0.3. software (AB Sciex, Stockholm, Sweden).

### 2.7. Metabolite Data

The generation of the metabolite data has been described in detail elsewhere [38]. In brief, fasting blood samples were stored in cold blocks and centrifuged within 30 min from the blood draw, whereafter plasma aliquots were immediately stored at −20 °C. Within one week, samples were transferred to −80 °C. During plasma sample preparation, proteins were precipitated by mixing with acetonitrile, followed by the steps of shaking, centrifugation, and filtering. Samples were analyzed in a UHPLC-qTOF-MS system in reversed-phase chromatography in positive and negative ionization modes.

### 2.8. Data Analysis

PCoA was used to visualize clustering of microbiota composition for the interventions (UniFrac distance metrics (unweighted and weighted)) with the R package Phyloseq (version 1.32.0). For supervised analysis, gut microbiota composition data was analyzed as relative abundance on genus level and filtered, such that each had to be present in at least 50% of the participants in any of the treatment or washout weeks. A pseudo count of 10^−3^ multiplied by the lowest value in the dataset was added to manage fold change calculations for zero values. To investigate potential effects of the diets on the gut microbiota composition, an in-house Random Forest (RF) algorithm was used (R package MUVR [57], version 4.0.0). This machine learning algorithm delivers a selection of the most important variables describing differences between intervention groups. The algorithm incorporates a double cross-validation design aimed at minimizing overfitting [57,58]. Due to dependency related to the crossover design, all analyses were performed as a pairwise multilevel analysis, conceptually similar to a paired *t*-test applied on multivariable data [59]. Results are presented as the classification rate (CR). Parameter settings for all MUVR models were nOuter = 0.8, nRep = 40 and varRatio = 0.85. The effect matrices created for the between-intervention multilevel models were created by the log fold change data of the FODMAP, gluten and placebo interventions. In addition, the within-intervention effects were investigated as log fold change for the respective intervention vs. their previous run-in or washout week. Permutation testing (*n* = 100) was performed to ensure that the results were not due to overfitting [57]. The parameters for the permutation tests were similar to those for the multilevel models. From models with p_permutation_ < 0.05 (FODMAPs vs. placebo, FODMAPs vs. washout and FODMAPs vs. gluten, henceforth referred to as ‘FODMAP-related models’), the genera with the highest predictive power were identified [57]. The Random Forest classification of the subtype was modelled with the end-of-intervention gut microbiota for FODMAPs and gluten. Metabolite features of interest were previously selected by RF models on the metabolomics data: FODMAPs vs. placebo and gluten vs. placebo [38]. To investigate possible interactions between genera and metabolites, similar to the above RF analysis, RF modelling of the combined metabolite and microbiota data was performed.

Selected gut microbiota genera of interest, SCFAs in feces and plasma and diversity measures were further investigated using a mixed model design with participant ID as a random factor and intervention and period as fixed factors, with type 3 tests and pairwise comparisons. The factor IBS subtype and the interaction IBS subtype x intervention were initially included, but did not contribute meaningfully to any of the models and were therefore removed from the final modelling. All variables for genera and SCFA were log-transformed and results back-transformed. For all models, homoscedasticity and normality were assessed by residual and quantile–quantile plots. Partial Spearman correlation analysis (adjusting for age and sex) was performed for each intervention with specific variables from genera, SCFAs and metabolome and the underlying variables in IBS-SSS, the bowel emptying diary and the health and quality-of-life questionnaires. Data were analyzed as end of intervention. SCFA cross-correlations and correlations with questionnaires included datapoints from both interventions and preceding washout weeks, adjusted for age, sex, and identity. The bowel-emptying-diary variable, ‘whether medication was needed for bowel emptying’, was excluded, due to severe zero inflation (ninety-one percent zeros).

Since the study was exploratory, there was no correction for multiple testing. However, due to the high number of tests in the correlation analysis, the stability of significant findings was evaluated according to the Benjamini–Hochberg false discovery rate procedure, considering the FDR-adjusted *p*-value < 0.05 as significant.

The study sample size was determined post hoc by assuming a difference of 50 points in the IBS-SSS with a power of 0.8, and a 20% dropout rate, using intra-individual SD for the interventions from this trial (SD = 111.6) [22]. The required sample size was 64 participants, with the level of significance in a two-sided test set to 0.05/3 (Bonferroni). All analyses were performed in the programming language R version 4.0.0. No adverse events were reported relating to the dietary interventions.

## 3. Results

Of 110 eligible participants, 6 were excluded either because they dropped out before visit 2 (*n* = 3) or visit 3 (*n* = 3), i.e., before entering any intervention arms. One subject dropped out after the first intervention (before visit 4) and did not return the questionnaires from that intervention (gluten). A flowchart is presented in Appendix A. Deviations from the protocol were: intake of probiotics (*n* = 1), antibiotics (*n* = 2), non-compliance (*n* = 7), deviations from inclusion criteria in BMI (*n* = 2), age (*n* = 2), lactose intolerance (*n* = 5), intake of symptom-mitigating pharmaceuticals (*n* = 2), and un-subtyped IBS (*n* = 14). Due to the exploratory nature of the study, all subjects were included in the present study except the ones consuming probiotics and antibiotics (*n* = 3). Hence, 100 subjects were included in the subsequent gut microbiota and plasma SCFA analyses (*n* = 100). Of these, 67 subjects had fecal SCFA samples from all three interventions, 19 from two interventions and 5 from one intervention. The corresponding numbers for the FODMAP, gluten, and placebo interventions were 82, 85 and 77 samples. Metabolomics data were missing for one subject in the FODMAP intervention due to a technical error during the instrument analysis. Otherwise, the data were complete (*n* = 100).

### 3.1. Gut Microbiota, SCFAs and Correlation with IBS Symptoms

The initial sequencing generated a median 137620 [SD 15471 (IQR 11950)] sequences per sample. After filtering and bioinformatic processing corresponding, values were 106062 [14907 (20236)] sequences. No intervention effect was found on gut microbiota diversity measured by richness, Shannon’s diversity index, Simpson’s index or inverse Simpson’s index (Table 1). Unsupervised PCoA did not show any separation between interventions based on gut microbiota composition, using either weighted or unweighted UniFrac (Appendix A). Supervised RF models showed differences in gut microbiota for FODMAP-related models (CR from 0.88 to 0.90, p_permutation_ ≤ 0.0001). However, no difference was observed between gluten and placebo interventions or with their respective washout weeks (CR ≤ 0.64, *p* ≥ 0.11; Appendix A). There was no evidence of specific IBS subtype effects in any of the analyses.

*Anaerostipes, Bifidobacterium, Faecalibacterium, Fusicatenibacter, Agathobacter, Paraprevotella*, and *Oxalobacter* increased after the FODMAP intervention compared to the placebo (*p* ≤ 0.01; Figure 1), whereas *Lachnoclostridium, Roseburia*, (*Ruminoccocus*) *torques*, *Lachnospiraceae* NK4A136, *Lachnospiraceae* NA, *Hungatella, Eisenbergiella, Negativibacillus, Coprostanoligenes* group NA, and *Flavinofractor* decreased (*p* ≤ 0.04; Figure 1). (Non-significant comparisons are presented in Appendix A).

Among the bacterial genera that increased after the FODMAP intervention compared to the placebo, *Bifidobacterium* correlated with pain linked to bowel emptying, *Faecalibacterium* correlated with SCBMs, *Agathobacter* correlated with bodily pain, and *Paraprevotella* correlated with dissatisfaction with bowel habits (r from 0.21 to 0.25, *p* ≤ 0.03), Figure 2. There was a pattern of some genera that correlated with frequency of abdominal pain and total IBS-SSS score (*Roseburia, Lachnoclostridium,* and an unknown *Lachnospiraceae* genus r ≤ 0.25, *p* < 0.05). However, those genera were not affected by FODMAP intake. None of the correlations remained statistically significant after adjustment for multiple testing.

Plasma isobutyrate was reduced after FODMAPs compared to the placebo (*p* = 0.02). Isovalerate in feces was reduced after gluten compared to the placebo (*p* = 0.01; Table 2). There were only weak correlations between the SCFAs in feces and in plasma. In particular, isovalerate in plasma correlated with isovalerate and isobutyrate in feces **(**Appendix A). There were also weak correlations between SCFAs (fecal caproate, propionate, acetate, butyrate, valerate, and succinate) and questionnaire data (severity of abdominal pain, number of bowel movements, SCBMs, interference with quality of life, and BSFS 6-7) (Appendix A).

### 3.2. Correlations between Bacterial Genera and Metabolites Related to FODMAP Exposure

Combining gut microbiota and metabolites in the Random Forest analysis provided a pronounced difference between the FODMAPs and placebo (CR = 0.91, p_permutation_
*<* 0.0001). Top selected variables of interest included *Anaerostipes*, *Bifidobacterium*, 3-indolepropionate (IPA), phenyl sulphates and as yet unidentified metabolites (Appendix A).

*Bifidobacterium* correlated positively with propionate in plasma, whereas several bacterial genera (*Negativibacillus, Turicibacter, Peptococcus,* an unknown *Christensenellaeae* genus, and an unknown *Coprostanotigenes* genus) correlated negatively with valerate, isovalerate, isobutyrate, succinate, acetate, butyrate, and propionate in feces (Appendix A).

*Agathobacter, Blautia, Anaerostipes, Butyricicoccus*, *Roseburia*, *Bifidobacterium*, and *Erysipelitracheae UCG*-003 correlated positively with phenolic-derived metabolites (3-3-hydroxyphenylpropionate, 3-hydroxyhippurate, and unidentified features), while *Flavinofractor*, *Eisenbergiella*, an unknown *Christensenellaceae* genus and *Peptococcus* correlated negatively. *Blautia*, *Anaerostipes*, *Fusicatenibacter*, *Butyriccoccus*, *Roseburia*, *Erysipelotrichaceae* UCG-003, the *Lachnospiraceae* ND3007_group, an unknown *Lachnospiraceae* genus, and *Turicibacter* correlated positively with IPA, an unknown quionoline/indole compound, and an unidentified feature, whereas *Flavinofractor*, an unknown *Christensenellaceae* genus, and an unknown *Coprostanotigenes* genus correlated negatively. *Marvinbryantia*, *Negativibacillus*, *Turicibacter*, *Eisenbergiella*, an unknown *Christensenellaceae* genus, and an unknown *Coprostanotigenes* genus, *Oxalobacter*, *Peptococcus*, and *Hungatella* correlated positively, whereas the *(Ruminoccocus)_grauveauii*_group, *Faecalibacterium*, *Agathobacter*, *Blautia*, *Anaerostipes*, *Fusicatenibacter*, *Butyricicoccus*, *Anaerovoracaceae* Family_XII_UCG-001, *Paraprevotella*, and the *Lachnospiraceae* NK4A136_group correlated negatively with unknown phenyl sulphates and a quinoline/indole derivate (Figure 3).

Phenolic-derived metabolites correlated with butyrate and valerate in plasma (Appendix A). Unknown phenyl sulphates correlated negatively with acetate, succinate, butyrate, and propionate in feces. There were only weak correlations between metabolites related to the gluten intervention and SCFAs (Appendix A).

## 4. Discussion

This is the first clinical trial to study changes in the gut microbiota composition, SCFAs, and the metabolome when provoking subjects with IBS with high doses of FODMAPs and gluten. FODMAPs, but not gluten, altered the gut microbiota composition, in particular causing an increased proportion of saccharolytic genera, corresponding to an increased exposure to fermentable carbohydrates. There were also minor effects of FODMAPs and gluten on SCFAs. FODMAP-related genera correlated with phenolic-derived metabolites and IPA. These metabolites have previously been associated with improved metabolic health and reduced risk of inflammation and diabetes type 2 [39,61,62]. There were only weak correlations of microbial genera and SCFAs with IBS symptoms. No outcomes differed among the IBS subtypes.

### 4.1. Intervention Effects on the Gut Microbiota Composition and SCFAs

The lack of difference in alpha diversity in our study was in accordance with previous results from interventions with low and high FODMAP intake [11,15,16]. The results are also in line with the view that FODMAPs may have a more modest effect on the gut microbiota composition than previously considered [16]. Our observation that *Bifidobacterium* increased after the FODMAP intervention is in accordance with other studies showing that this genus is, in fact, most consistently associated with FODMAP intake [15,16]. The increase of the saccharolytic genera *Anareostipes, Faecalibacterium, Fusicatenibacter, Agathobacter,* and *Paraprevotella* after the FODMAP intervention has not been consistently reported in IBS and FODMAP trials, although they have previous been increased in some trials with exposure to FODMAPs, plant foods and dietary fiber [16,63,64,65,66].

The fact that *Lachnoclostridium, Lachnospiraceae* NK4A136, *Hungatella*, and *Negativibacillus* decreased after the FODMAP intervention is also supported by previous results [67,68,69,70], although plausible mechanisms are still lacking [71]. *Roseburia* has been reported both to decrease and increase after dietary fiber-rich whole grain intake [72]. Similar to our study, the *Roseburia* genus was previously reported to be suppressed after high intake of FODMAPs [73]. In line with a previous trial on high-fiber rye intake [64], *(Ruminoccocus) torques* was reduced after the FODMAP intervention. This genus has been associated with increased gut permeability and inflammation [74,75,76,77]. Interestingly, the observed reduction of *Flavinofractor* by FODMAPs in this study is in line with a previously reported reduction in relation to plant foods [78]. Moreover, this genus was found to be enriched in IBS [79,80], and although it was previously associated with abdominal pain in subjects with IBS [81], *Flavinofractor* was not associated with IBS symptoms in this study. The mechanistic role of *Flavinofractor* in relation to health, both in subjects with and without IBS, remains unclear [82], and requires further investigation.

Previous studies with low and high intake of FODMAPs have only had a minor or even a lack of effect on SCFAs in feces [12,15,16]. In our study, the lower level of isobutyrate in plasma after FODMAP intake was in accordance with gut microbiota favoring carbohydrates as a substrate, effectively reducing the fermentation of proteins into branched-chain fatty acids [82]. The lack of a more general effect of FODMAPs on SCFAs is assumed to relate to fermentation occurring in the proximal colon, making feces a suboptimal matrix [17]. However, these issues do not address the minor effect observed in plasma. Taken together, FODMAP intake had only a minor effect on both fecal and plasma SCFAs, concordant with the literature [12,15,16].

### 4.2. Correlations between Gut Microbiota and IBS Symptoms

*Bifidobacterium* correlated weakly with pain linked to bowel emptying, although it had previously not been related to IBS symptoms [11,12,14]. Otherwise, there was no clear pattern among the correlations between microbiota and IBS symptoms, thus likely representing spurious associations. In general, there is limited consistency among studies, both regarding genera affected by FODMAP intake and their effect on IBS symptoms. This may relate to several challenges, such as inconsistencies in methods for analyzing gut microbiota [83], IBS being a heterogenous condition [84] with high placebo response [85], and the fact that trials have been of small sample size [6]. In addition, feces samples are a heterogenous material, and challenges remain to ensure accurate measurements [86].

### 4.3. Correlations between Bacterial Genera and Metabolites Related to Intake of FODMAPs

It was previously shown that the conversion of polyphenols to intermediate metabolites, such as 3-3-hydroxyphenylpropionate and 3-hydroxyhippurate, depends on gut microbiota [87,88,89]. In our study, *Agathobacter* increased with FODMAPs and correlated most strongly with phenolic derivates, which to our knowledge has not previously been reported. Interestingly, similar to our study, the *Bifidobacterium* and *Fusicatenibacter* species were reported to correlate with 3-3-hydroxyphenylpropionate and 3-hydroxyhippurate in an online atlas of human plasma microbial-derived metabolites [36]. However, to gain an understanding of the mechanisms involved, more efforts are required to investigate the involvement of microbial conversions [88].

Several genera that increased after FODMAP intake correlated with IPA and a quinoline/indole derivative in plasma. Among these genera, *Anaerostipes* and *Fusicatenibacter* have previously been associated with IPA [36]. It is known that IPA is produced by microbiota, but more research is needed for deeper understanding of which microbiota are involved and their mechanistic routs [90]. Moreover, FODMAP intake reduced the level of hyodexoycholic acid, which is a secondary bile acid produced by gut microbiota [33]. However, the proportion of hyodeoxycholic acid-producing bacteria was not influenced by FODMAPs exposure, suggesting a direct beneficial effect of FODMAP intake per se [91]. Only *Oxalobacter* correlated with the group of phenyl sulfates and unidentified features affected by FODMAP intake. Identification of these features and their interplay with gut microbiota needs to be further studied.

### 4.4. Gluten

It is well known that carbohydrates are preferred substrates for gut microbiota fermentation, whereas protein fermentation normally occurs when carbohydrates are depleted [9]. Even though we provoked them with high doses of gluten, participants simultaneously consumed carbohydrates in their diet. The lack of effect of the gluten intervention on alpha diversity and gut microbiota composition was consequently not surprising, and consistent with previous studies [26].

Concerning SCFAs, the reduced level of isovalerate in plasma after the gluten intervention was surprising: isovalerate is a branched-chain fatty acid, and if anything, increased levels from protein breakdown would be expected [92]. Interestingly, there was a weak trend of all fecal SCFA being lower after gluten intake, compared to the other interventions, although only isovalerate reached significance. A previous study investigating the effect of a gluten-free diet in healthy adults of a low- (2 g/day) and high- (18 g/day) gluten diet did not observe an effect on SCFAs in feces or plasma [26]. Another study provoked participants with 30 g gluten for 4 days and found increased SCFAs in feces, although the authors note that the gluten fraction used also contained a small amount of non-absorbable starch [28]. Taken together, the observed effect on isovalerate may be spurious, but it is also possible that gluten had a general lowering effect on SCFAs, with a lack of mechanistic explanation.

### 4.5. Limitation and Strength

This trial has several limitations. The intervention periods were of relatively short duration, and the effects on the microbiota and metabolites could have been more pronounced with longer durations. However, the intervention periods were decided on to enhance participation for the primary endpoint of investigating IBS symptoms and the microbiota investigation was exploratory. Regardless, the effect after one week of intervention are still of high interest. Similarly, the one-week washouts might not have been sufficient to eliminate carry-over. However, analysis of the sequence order did not show any indications of systematic differences. We therefore assume that one week of washout was sufficient. Furthermore, a hydrogen breath test was not used, which would have given a direct measure of fermentation. The use of sucrose as a placebo can be questioned, due to reported sucrase-isomaltase deficiency in IBS [93]. However, since there was no effect of placebo in this trial (established by comparing it to the washout weeks), the choice of sucrose seems to have had a negligible impact on the results. Moreover, feces samples were freeze dried for the analysis of SCFAs. For some samples there were not enough material; therefore, the number of samples is lower for the analysis of SCFAs in feces compared to other biological data. Lastly, one must remember that there are several challenges with microbiota analysis. Until today the profile of a ‘normal’ microbiota is unknown, and it probably relates to context [94,95]. Importantly, there are several methodological challenges with microbiota analysis and disparities in how to measure microbiota among studies [83]. In the present study, gut microbiota composition was assessed using 16S rRNA gene sequencing, whereas shotgun metagenomics could have contributed more information on gut microbiota functionality and activity, which are known to differ between species and strains [29,32,96]. The challenges of microbiota analysis, together with the heterogenicity of IBS [94], large placebo responses [85], and differences in study design [6] could relate to the large diversity reported in clinical studies.

A particular strength of this study was the double-blind study design, which reduced the risk of bias, combined with a large sample size and a three-way placebo-controlled cross-over design, which should limit the contribution of individual variability to treatment effects. Another strength is the elimination-exposure character of the study, making the molecular responses specific to the exposures, not confounded by the treatment effect. Furthermore, the joint analysis of gut microbiota and metabolites added knowledge about the physiological effect of FODMAP intake, both regarding gut microbiota composition and functionality, which may be of relevance to health, although more research is needed.

## 5. Conclusions

Our combined gut microbiota- and metabolite-centered approach showed that the intake of FODMAP, but not gluten, over one week altered the gut microbiota composition, which correlated with metabolites positively associated with health. Specifically, the genera *Agathobacter, Anaerostipes, Fucicatenibacter,* and *Bifidobacterium* correlated with increased plasma concentrations of phenolic-derived metabolites and 3-indolepropionate. The weak correlations of FODMAP-related gut microbiota and metabolites with IBS symptoms indicate that responses to the interventions had limited impact on symptoms. Consequently, the minor effect of FODMAPs on IBS symptoms must be weighed against beneficial health effects. However, since the interventions were of short duration, the results should be interpreted with caution and long-term randomized controlled trials are highly warranted.

## Figures and Tables

**Figure 1 nutrients-15-03045-f001:**
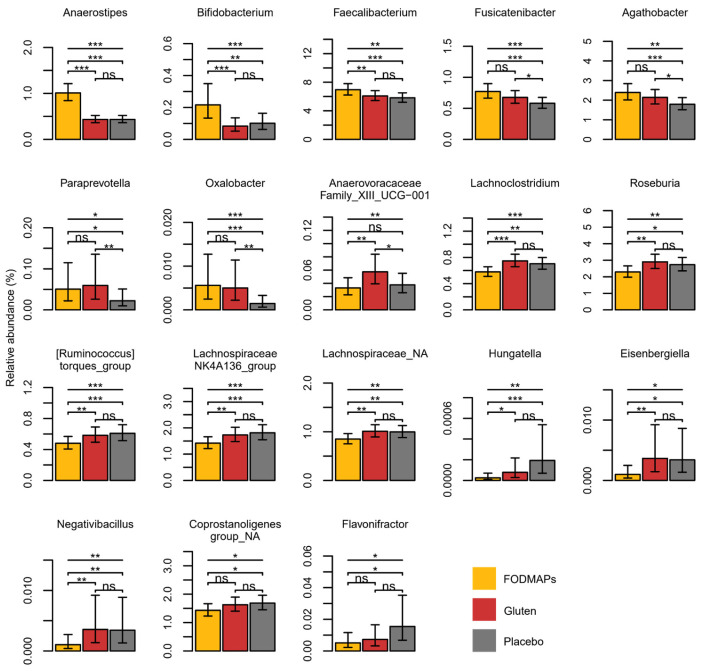
Bacterial genera selected from Random Forest modelling for FODMAP-related models (i.e., FODMAPs vs. placebo, FODMAPs vs. washout and FODMAPs vs. gluten). Genera (relative abundance (%)) differing between the interventions of FODMAPs, gluten and placebo in mixed models are presented (n = 100). Data presented as estimated marginal means and 95% confidence interval. The top row in each graph represents the overall *p*-value, while the others present pairwise comparisons. *p*-values are presented with the star system: * = *p* < 0.05, ** = *p* < 0.01, *** = *p* < 0.001. The figure is reproduced from [60]. Abbreviation: FODMAPs, fermentable oligosaccharides, disaccharides, monosaccharides, and polyols; ns: non significant.

**Figure 2 nutrients-15-03045-f002:**
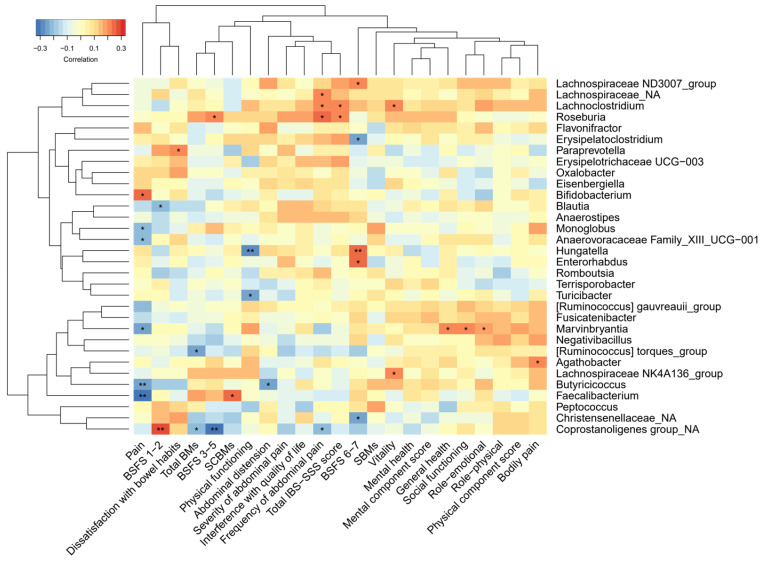
Partial Spearman correlation between bacterial genera selected from Random Forest modelling for FODMAP-related models (FODMAPs vs. placebo, FODMAPs-washout and FODMAPs vs. gluten) and the questionnaires IBS-SSS, Short Form 36 version 2 (health and quality of life) and the bowel diary, adjusted for age and sex (n = 100). The figure is reproduced from [60]. *p*-values are presented with the asterisk system: * = *p* < 0.05, ** = *p* < 0.01. Abbreviations: BM, Bowel movements; BSFS, Bristol Stool Form Scale; FODMAPs, fermentable oligosaccharides, disaccharides, monosaccharides, and polyols; IBS-SSS, Irritable Bowel Syndrome Severity Scoring System; SBMs, spontaneous bowel movement; SCBMs, spontaneous complete (a sensation of complete evacuation) bowel movement; Pain, abdominal pain linked to bowel emptying.

**Figure 3 nutrients-15-03045-f003:**
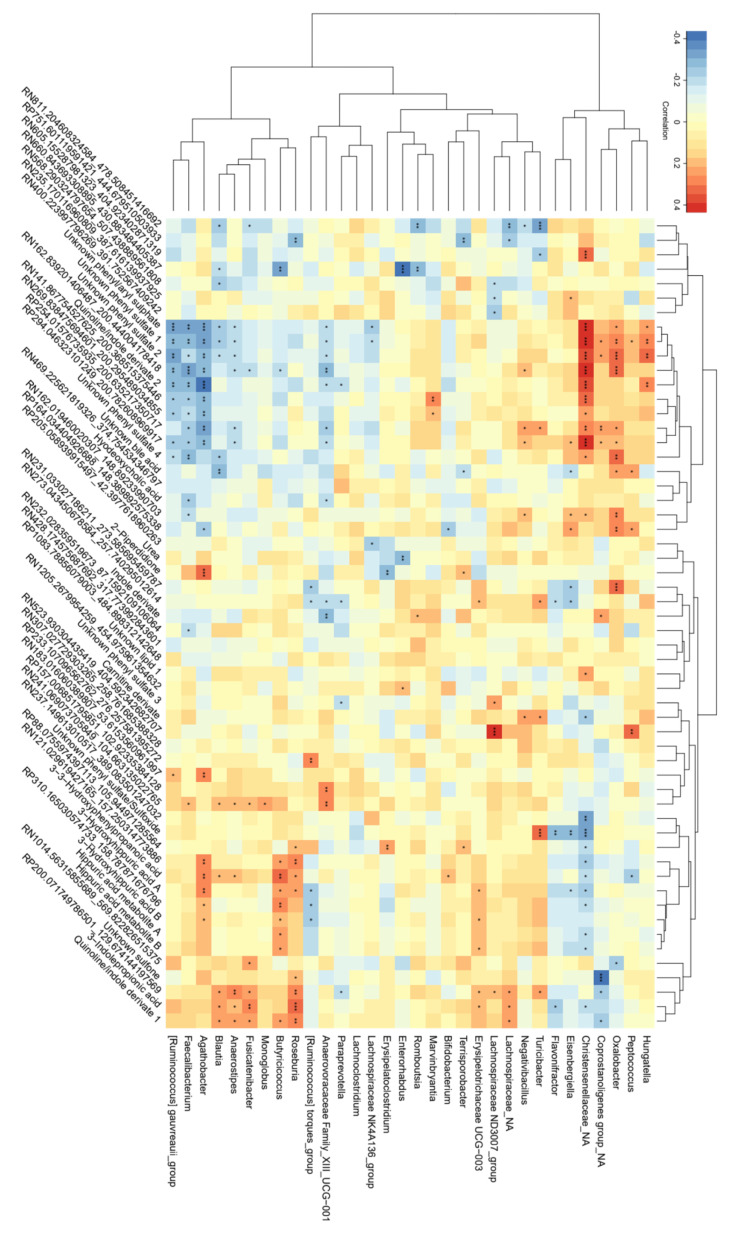
Partial Spearman correlation between genera selected from Random Forest modelling for FODMAP-related models (i.e., FODMAPs vs. placebo, FODMAPs vs. washout and FODMAPs vs. gluten) and metabolites selected from FODMAP vs. placebo, adjusted for age and sex (n = 99). *p*-values are presented with the asterisk system: * = *p* < 0.05, ** = *p* < 0.01, *** = *p* < 0.001. The figure is reproduced from [60]. Abbreviation: FODMAPs, fermentable oligosaccharides, disaccharides, monosaccharides, and polyols.

**Table 1 nutrients-15-03045-t001:** Alpha diversity measures for the interventions FODMAPs, gluten and placebo (*n* = 100).

	FODMAPs	Gluten	Placebo	*p*-Value
Richness	436.77 (421.6–451.94)	447.93 (432.76–463.1)	447.14 (431.97–462.31)	0.10
Shannon’s diversity index	4.59 (4.51–4.66)	4.63 (4.56–4.7)	4.63 (4.56–4.71)	0.20
Simpson’s diversity index	0.97 (0.97–0.98)	0.97 (0.97–0.98)	0.97 (0.97–0.98)	0.76
Inv Simpson’s diversity index	46.84 (42.78–50.89)	47.83 (43.78–51.89)	47.92 (43.87–51.98)	0.81

Abbreviation: FODMAPs, fermentable oligosaccharides, disaccharides, monosaccharides, and polyols.

**Table 2 nutrients-15-03045-t002:** Short-chain fatty acids (SCFAs) in feces (*n* = 91) and plasma (*n* = 100) after the FODMAP, gluten and placebo intervention. Data are expressed as estimated marginal means and 95% confidence interval. Pairwise comparisons differing at *p* < 0.05 are presented with lettering.

SCFA	FODMAPs	Gluten	Placebo	*p*-Value
FECES (μmol/g- dry weight)				
Acetate	85.82 (74.3–99.14)	78.6 (68.16–90.65)	83.53 (72.09–96.8)	0.46
Propionate	28.45 (24.71–32.76)	25.78 (22.42–29.63)	27.57 (23.87–31.84)	0.35
Butyrate	34.8 (29.08–41.64)	31.26 (26.17–37.43)	33.91 (28.24–40.73)	0.43
Isobutyrate	2.27 (1.85–2.78)	1.96 (1.6–2.39)	2.55 (2.06–3.14)	0.18
Succinate	1.3 (1.01–1.67)	1.03 (0.81–1.33)	1.06 (0.82–1.37)	0.23
Valerate	4.45 (3.94–5.03)	4.17 (3.7–4.7)	4.65 (4.1–5.27)	0.28
Isovalerate	2.47 ^b^ (2.21–2.77)	2.09 ^a^ (1.87–2.34)	2.52 ^b^ (2.25–2.84)	0.01
Caproate	0.88 (0.63–1.22)	0.83 (0.60–1.14)	0.96 (0.69–1.33)	0.57
PLASMA (µmol/L)				
Formate	195.37 (175.81–217.11)	195.71 (176.11–217.49)	196.4 (176.74–218.25)	0.99
Acetate	56.39 (49.9–63.72)	57.77 (51.12–65.29)	61.02 (53.99–68.96)	0.47
Propionate	0.72 (0.58–0.9)	0.69 (0.56–0.85)	0.75 (0.61–0.92)	0.72
Butyrate	0.36 (0.3–0.43)	0.34 (0.28–0.41)	0.32 (0.27–0.39)	0.53
Isobutyrate	0.21 ^a^ (0.18–0.24)	0.24 ^b^ (0.21–0.27)	0.25 ^b^ (0.22–0.28)	0.04
Succinate	3.09 (2.87–3.33)	3.12 (2.89–3.36)	3.17 (2.94–3.41)	0.75
Valerate	0.05 (0.04–0.06)	0.05 (0.04–0.06)	0.06 (0.05–0.07)	0.42
Isovalerate	0.19 (0.16–0.23)	0.19 (0.16–0.23)	0.22 (0.19–0.27)	0.12
Caproate	0.14 (0.12–0.17)	0.15 (0.12–0.17)	0.16 (0.13–0.19)	0.45

Abbreviation: FODMAPs, fermentable oligosaccharides, disaccharides, monosaccharides, and polyols.

## Data Availability

The data is not publicly available, due to privacy restrictions. The data presented in this study are available on request from the corresponding author.

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
