# Peer review of "Effects of FODMAPs and Gluten on Gut Microbiota and Their Association with the Metabolome in Irritable Bowel Syndrome: A Double-Blind, Randomized, Cross-Over Intervention Study"

_nutrients, 2023, doi:10.3390/nu15133045_

Round 1

Reviewer 1 Report

The manuscript title should be rephrased to reflect the work performed. Please do not make an answer or question in the title of any scientific manuscript. 

The abstract should be rewritten to ensure there is coherence between the background, aims and the highlighted results. In addition, the English of the manuscript requires some improvement as syntax errors are found throughout the manuscript. Some of the sentences are either difficult to understand or meaningless. 

Gut microbiota profiles differentiated FODMAPs versus gluten and placebo, whereas gluten did not differentiate (lines 22-23) - What is the meaning? Most statistical outputs need to be rewritten.

What is the rationale the study single out gluten? There must be explanation on the implication of the current work to the real food consumption patterns.  

From the last paragraph of the introduction, it seems some past studies have been performed by many research groups and the results from the present study seem well expected. What is new in the present study? This paragraph should be rewritten to highlight the problem statement/research question before the aim of the present study. 

In the dietary intervention, from where FODMAPs is obtained? What is the source of FODMAPs? Why the placebo is using sucrose which may create other complication?  

Some of key information on the dietary intervention should be provided in the manuscript instead of referring to the past publication, e.g. how long is the duration of dietary intervention? The intervention program should be fully described.

How many fecal samples were obtained from each respondent under the dietary program? 

Highlight the method/instrument used for SCFA determination. 

A substantial amount of data were obtained. Thus, results display in tables and figures should be highlighted and discussed instead of outlining results from supplementary materials (tables and figures). The key results obtained should be emphasized in the manuscript. 

Please standardize the p-value indicating significant difference to p<0.05 or p<0.001.

For any correlation coefficient less than 0.5 (r<0.5), it is meaningless to mention despite it is significant. Please highlight those with strong correlation (r>0.5)   

Since there is no clear information could be obtained from  the correlation, more in depth relationship such as multivariate statistics should be performed for a better explanation. 

The discussion should be focused on the results obtained and explain the findings (new information) from the body of knowledge in the field. It should not limited to support the past studies. 

Conclude based on the findings obtained from the current study. Avoid having explanations or speculations which should be written in the discussion. What is the implication of the finding to the current knowledge in the field. 

The quality of English is inconsistent, some parts seem acceptable but other parts with many errors.

Author Response

We highly appreciate the time taken by the reviewer, and the valuable feedback they have provided. We have responded to each comment point by point. When referring to pages and lines in the manuscript, it refers to the revised manuscript with all changes accepted.

 Reviewer 1

Comments and Suggestions for Authors

The manuscript title should be rephrased to reflect the work performed. Please do not make an answer or question in the title of any scientific manuscript. 

Thank you for the comment, the revised title: ’Effects of FODMAPs and gluten on gut microbiota and their association with the metabolome in irritable bowel syndrome, a double-blind randomized cross-over intervention study’

’The abstract should be rewritten to ensure there is coherence between the background, aims and the highlighted results. In addition, the English of the manuscript requires some improvement as syntax errors are found throughout the manuscript. Some of the sentences are either difficult to understand or meaningless. 

Thank you for the comment, indeed, the abstract needed clarifications. The abstract has been changed to:

Background: A mechanistic understanding of the effects of dietary treatment in irritable bowel syndrome (IBS) is lacking. Our aim was therefore to investigate how fermentable oligo- di-, monosaccharides, and polyols (FODMAPs) and gluten affected gut microbiota and circulating metabolite profiles as well as to investigate potential links between gut microbiota, metabolites, and IBS symptoms. Methods: We used data from a double-blind, randomized, crossover study with week-long provocations of FODMAPs, gluten, and placebo in participants with IBS. To study the effects of the provocations on fecal microbiota, fecal and plasma short-chain fatty acids, the untargeted plasma metabolome, and IBS symptoms, we used Random Forest, linear mixed model and Spearman correlation analysis. Results: FODMAPs increased fecal saccharolytic bacteria, plasma phenolic-derived metabolites, 3-indolepropionate, and decreased isobutyrate and bile acids. Gluten decreased fecal isovalerate and altered carnitine derivatives, CoA, and fatty acids in plasma. For FODMAPs, modest correlations were observed between microbiota and phenolic-derived metabolites and 3-indolepropionate, previously associated with improved metabolic health and reduced inflammation. Correlations between molecular data and IBS symptoms were weak. Conclusion: FODMAPs, but not gluten, altered microbiota composition and correlated with phenolic-derived metabolites and 3-indolepropionate, with only weak associations to IBS symptoms. Thus, the minor effect of FODMAPs on IBS symptoms must be weighed against the effect on microbiota and metabolites related to positive health factors.

Gut microbiota profiles differentiated FODMAPs versus gluten and placebo, whereas gluten did not differentiate (lines 22-23) - What is the meaning? Most statistical outputs need to be rewritten.
Thank you for the comment, the abstract has been adjusted, see above

What is the rationale the study single out gluten? There must be explanation on the implication of the current work to the real food consumption patterns.  

Thank you for the comment. FODMAPs is the dietary constituent with the largest body of evidence for an effect on IBS. However, at the time of planning our study the effect of gluten on IBS was heavily discussed since previous studies were inconclusive. Hence, there was a need for more research, and gluten was therefore included as a study arm.

From the last paragraph of the introduction, it seems some past studies have been performed by many research groups and the results from the present study seem well expected. What is new in the present study? This paragraph should be rewritten to highlight the problem statement/research question before the aim of the present study. 

Thank you for the comment. The following changes have been made in the introduction:

In the manuscript (page 2, lines 55-57): FODMAPs have consistently been found to increase Bifidobacterium [11–15], but such effects have not been related to the severity of IBS symptoms [11,12,14]. Added in manuscript  page 2, lines 57-58: Other effects of FODMAPs on the gut microbiota have been inconclusive [16].

In the manuscript (page 2, lines 66-68): Interestingly, a study on healthy subjects consuming a diet rich in gluten found no effects on the gut microbiota composition [26]. Added in manuscript page 2, lines 68: ‘However, no such study has been performed in people with IBS.

Added in manuscript page 3, lines 100-104: ‘Since there are inconsistencies in the effects of FODMAPs and gluten on microbiota and SCFAs, and there is also a general lack of multiomics analysis in IBS and dietary trials, the aim of the present exploratory work was to investigate the effects of FODMAPs, gluten or placebo on both gut microbiota composition, fecal and plasma SCFAs and their relationship to IBS symptoms and the metabolome.

By these changes we hope to have clarified the novelty in our study.

In the dietary intervention, from where FODMAPs is obtained? What is the source of FODMAPs? Why the placebo is using sucrose which may create other complication?  

The interventions foods are described in section 2.2 ’Dietary interventions’. Further details about the foods are presented in Supplementary Table 1 and 2. We have added that the interventions were served as powders, page 4, line 148: ‘in powder form’

We agree with the reviewer regarding sucrose. In fact, if we could have re-performed the trial, we would instead have used maltodextrin as a control. However, since there was no effect of placebo in this trial (by comparing to the wash-out weeks), the choice of sucrose seems to have had a negligible impact on the results. This limitation is presented in the first publication of this study (https://pubmed.ncbi.nlm.nih.gov/34617561/) and was additionally added in the current manuscript page 16, line 555-558: ‘The use of sucrose as placebo can be questioned due to reported sucrase-isomaltase deficiency in IBS [98]. However, since there was no effect of placebo in this trial (by comparing to the wash-out weeks), the choice of sucrose seems to have had a negligible impact on the results.’

Some of key information on the dietary intervention should be provided in the manuscript instead of referring to the past publication, e.g. how long is the duration of dietary intervention? The intervention program should be fully described.
Thank you for the comment. We have made clarification in page 3 lines 122-129 it is stated that ‘During the seven weeks of the study, participants consumed a so-called low-impact diet, i.e. excluding gluten and having minimal consumption of FODMAPs, guided by a dietitian. Between the two initial run-in weeks on low-impact diet, a single combined challenge test with FODMAPs and gluten was carried out, following which blood samples were drawn at regular intervals during four hours for later analysis. Thereafter, participants underwent one-week interventions with FODMAPs, gluten and placebo, respectively, with one wash-out week in-between’.’ The study design is visualized in Supplementary Figure 1.

How many fecal samples were obtained from each respondent under the dietary program? 

In page 3 lines 129-131 it is stated that: ‘At the end of each week 2-7, participants returned questionnaires along with a feces sample, anthropometric measures were registered, and fasting blood samples were drawn, reflecting exposures during each previous week.’ Hence, participants handed in 6 fecal samples during the study. In addition, in the result section (page 7 lines 340-342) it is clarified how many participants actually handed in fecal samples from three/two/one interventions and distributed across each intervention: ‘Of these, 67 subjects had fecal SCFA samples from all three interventions, 19 from two interventions and 5 from one intervention. Corresponding number for the FODMAP, gluten, and placebo interventions were 82, 85 and 77 samples.’

Highlight the method/instrument used for SCFA determination. 

The method section including details about the SCFA analysis previously in supplementary is now within the manuscript, section 2.6.

A substantial amount of data were obtained. Thus, results display in tables and figures should be highlighted and discussed instead of outlining results from supplementary materials (tables and figures). The key results obtained should be emphasized in the manuscript.

We agree, the present manuscript includes a large amount of data. The table showing the result from the alpha diversity analysis has been moved from the supplementary material to the manuscript, effectively keeping the discussion focused on tables and graphs within the main manuscript.

Please standardize the p-value indicating significant difference to p<0.05 or p<0.001.

In the tables presenting correlations the significance level is presented as * = p<0.05, ** = p<0.01, *** = p<0.001. For clarity, in the text the exact p-value is presented, which we consider accurate.

For any correlation coefficient less than 0.5 (r<0.5), it is meaningless to mention despite it is significant. Please highlight those with strong correlation (r>0.5)   
The colorkeys in the heatmaps show there is no correlation >0.5, the scales go up to 0.4 in these analyses. We disagree that a correlation less than 0.5 is meaningless. In fact, weaker associations (such as frequently occurring from dietary exposures) can be highly interesting even though underlying associations may be diluted by other determinants. The manuscript states that this is an exploratory analysis which should be further evaluated in future studies.

Since there is no clear information could be obtained from  the correlation, more in depth relationship such as multivariate statistics should be performed for a better explanation. 
In this study, the most informative variables of microbiota for the interventions were in fact selected with machine learning modelling with random forest (i.e., multivariate analysis). Correlation analyses were used to further visualize and interpret how these genera associated to SCFAs, metabolites from the untargeted metabolomics analysis, and IBS symptoms. The weak associations are interesting per se, which could relate to that the intervention were only ongoing for one week, as discussed in the limitation section page 15-16 lines 546-551.

The discussion should be focused on the results obtained and explain the findings (new information) from the body of knowledge in the field. It should not limited to support the past studies. 

We thank the reviewer for the comment. We have condensed the discussion a bit (Previous submitted manuscript page 12, line 408-413, 423-429) but in general we see a great value to compare our findings with previous data, since the number of studies in this area is rather few.

Removed text:

Christensenellaeae has previously been associated with intake of galacto-oligosaccarides [78], and is generally considered inversely correlated to BMI [79]. A lack of effect in our trial could be due to the short timeframe of the study.  It should also be noted that although changes in proportion of genera may be a direct effect of the intervention, it cannot be ruled out that it may also result as an indirect effect due to different proportions of other affected genera.

Of note is that the interventions were restricted to one week. It is possible there would have been more pronounced effects on the gut microbiota, SCFAs, and the metabolome if the interventions were longer. Few studies have investigated the effect of FODMAP exposure on the gut microbiota and SCFA for a longer duration. One study found no effect on SCFA after 10 days of provocation with a daily dose of 16 g fructooligossacharides [12]. Hence, future trials should investigate the effect on gut microbiota and metabolites by FODMAPs and gluten for a longer time.

Conclude based on the findings obtained from the current study. Avoid having explanations or speculations which should be written in the discussion. What is the implication of the finding to the current knowledge in the field. 

Thank you for the suggestion. The conclusion has been reduced:

Section 5, page 16 lines 570-580: Our combined gut microbiota and metabolite-centered approach showed that the in-take of FODMAP, but not gluten, over one week altered the gut microbiota composition which correlated with metabolites positively associated to health. Specifically, the genera Agathobacter, Anaerostipes, Fucicatenibacter, and Bifidobacterium correlated with increased plasma concentrations of phenolic-derived metabolites and 3-indolepropionate.The weak correlations of FODMAP-related gut microbiota and metabolites with IBS symptoms in-dicate that responses to the interventions had limited impact on symptoms. Consequently, the minor effect of FODMAPs on IBS symptoms must be weighed against beneficial health effects. However, since the interventions were of short duration, the results should be in-terpreted with caution and long-term randomized controlled trials are highly warranted.

Comments on the Quality of English LanguageThe quality of English is inconsistent, some parts seem acceptable but other parts with many errors.

The manuscript has now been proofread to correct the quality of the English

Reviewer 2 Report

A well written exploratory analysis of the previously published RCT. Interesting to read and leaves little room for improvement. Congratulations for a solid paper. 

Minor comments. 

- short duration (1 week) needs to be added into the abstract. 

- authors might want to discuss the role of Flavonifactor in IBS, as FODMAPs seem to have an effect on it. Enrichment of Flavonifractor may me associated with IBS. See for example. 
https://pubmed.ncbi.nlm.nih.gov/37173627/
https://www.ncbi.nlm.nih.gov/pmc/articles/PMC6504675/ 
- despite the results of the analysis, only long-term RCTs can ultimately verify the overall health effects of low-FODMAP diet; this should mentioned
- The conclusions paragraph includes some vague text, the exact meaning of some sentences remain unclear. Re-writing the conclusions is suggested.  

Some minor errors and vague text lines

Author Response

We highly appreciate the time taken by the reviewer, and the valuable feedback they have provided. We have responded to each comment point by point. When referring to pages and lines in the manuscript, it refers to the revised manuscript with all changes accepted.

Reviewer 2

Comments and Suggestions for Authors:

A well written exploratory analysis of the previously published RCT. Interesting to read and leaves little room for improvement. Congratulations for a solid paper. 

Thank you!

Minor comments. 

- short duration (1 week) needs to be added into the abstract. 
We agree, this should be included. The abstract has been changed to:

Background: A mechanistic understanding of the effects of dietary treatment in irritable bowel syndrome (IBS) is lacking. Our aim was therefore to investigate how fermentable oligo- di-, monosaccharides, and polyols (FODMAPs) and gluten affected gut microbiota and circulating metabolite profiles as well as to investigate potential links between gut microbiota, metabolites, and IBS symptoms. Methods: We used data from a double-blind, randomized, crossover study with week-long provocations of FODMAPs, gluten, and placebo in participants with IBS. To study the effects of the provocations on fecal microbiota, fecal and plasma short-chain fatty acids, the untargeted plasma metabolome, and IBS symptoms, we used Random Forest, linear mixed model and Spearman correlation analysis. Results: FODMAPs increased fecal saccharolytic bacteria, plasma phenolic-derived metabolites, 3-indolepropionate, and decreased isobutyrate and bile acids. Gluten decreased fecal isovalerate and altered carnitine derivatives, CoA, and fatty acids in plasma. For FODMAPs, modest correlations were observed between microbiota and phenolic-derived metabolites and 3-indolepropionate, previously associated with improved metabolic health and reduced inflammation. Correlations between molecular data and IBS symptoms were weak. Conclusion: FODMAPs, but not gluten, altered microbiota composition and correlated with phenolic-derived metabolites and 3-indolepropionate, with only weak associations to IBS symptoms. Thus, the minor effect of FODMAPs on IBS symptoms must be weighed against the effect on microbiota and metabolites related to positive health factors.

- authors might want to discuss the role of Flavonifactor in IBS, as FODMAPs seem to have an effect on it. Enrichment of Flavonifractor may me associated with IBS. See for example.

Thank you for the references!  Flavonifractor is now further discussed in the paper, page 14 lines 475-481:
Interestingly, the observed reduction of Flavinofractor by FODMAPs in this study is in line with a previously reported reduction in relation to plant foods [79]. Moreover, this genus was found to be enriched in IBS [80,81], and although it was previously associated to abdominal pain in subjects with IBS [82], Flavinofractor did not associate to IBS symptoms in this study. The mechanistic role of Flavinofractor in relation to health both in subjects with and without IBS remains unclear [83] and requires further investigation.

- despite the results of the analysis, only long-term RCTs can ultimately verify the overall health effects of low-FODMAP diet; this should mentioned
Thank you for highlighting this important point. The following sentence has been added to the conclusion (page 16, line 579-580): ‘However, since the interventions were of short duration, the results should be interpreted with caution and long-term randomized controlled trials are highly warranted.’

- The conclusions paragraph includes some vague text, the exact meaning of some sentences remain unclear. Re-writing the conclusions is suggested.  

Thank you for the comment. The conclusion has been reduced (page 16, lines 570-580):

Our combined gut microbiota and metabolite-centered approach showed that the in-take of FODMAP, but not gluten, over one week altered the gut microbiota composition which correlated with metabolites positively associated to health. Specifically, the genera Agathobacter, Anaerostipes, Fucicatenibacter, and Bifidobacterium correlated with increased plasma concentrations of phenolic-derived metabolites and 3-indolepropionate. The weak correlations of FODMAP-related gut microbiota and metabolites with IBS symptoms indicate that responses to the interventions had limited impact on symptoms. Consequently, the minor effect of FODMAPs on IBS symptoms must be weighed against beneficial health effects. However, since the interventions were of short duration, the results should be interpreted with caution and long-term randomized controlled trials are highly warranted.

Comments on the Quality of English Language: Some minor errors and vague text lines

The manuscript has now been proofread to correct the quality of the English

Reviewer 3 Report

The authors propose an interesting work, as sometimes privative diets are proposed too easily and without a scientific rationale

- The main limitation is due to the microbiota, which seems to be the center of the work; the Bristol scale for feces has not been considered; this certainly influences the final result, just as the strains can be variable according to the subjects.

- There are no changes in quality of life or outcomes due to IBS, bowel regularity, stool consistency, or sensations in general.

- The nutritional scheme adopted is not clear, a typical day should be clearly shown and how the plan was administered (generic indications or detailed program); nothing is said about the use of supplements

- Nothing is said about physical activity

- Possible food intolerances independent of IBS, which should be investigated before the intervention, should be considered

it needs some revision

Author Response

We highly appreciate the time taken by the reviewer, and the valuable feedback they have provided. We have responded to each comment point by point. When referring to pages and lines in the manuscript, it refers to the revised manuscript with all changes accepted.

Reviewer 3

The authors propose an interesting work, as sometimes privative diets are proposed too easily and without a scientific rationale

- The main limitation is due to the microbiota, which seems to be the center of the work; the Bristol scale for feces has not been considered; this certainly influences the final result, just as the strains can be variable according to the subjects.

Thank you for the comment. The aim with this work was to study the effects of FODMAPs, gluten or placebo on both gut microbiota composition, fecal and plasma SCFAs and their relationship to IBS symptoms and the metabolome. There were only minor effects on SCFAs, therefore the manuscript appears to focus on microbiota. We are aware that the passage time is a major implication for the microbiota composition but we have not considered that in our study design since it was an exploratory study and there was little knowledge about this when the present study was designed. It is also unclear how well Bristol score can capture the passage time to be used as a tool/confounder in the models. We recently published a discrepancy between subjectively reported Bristol stool form scale and objectively measured water content (https://pubmed.ncbi.nlm.nih.gov/36087104/), highlighting a challenge using subjectively reported Bristol scale.

- There are no changes in quality of life or outcomes due to IBS, bowel regularity, stool consistency, or sensations in general.

It is correct that these are the findings in our study after exposure of FODMAPs, gluten and placebo, reported in a previous publication (https://pubmed.ncbi.nlm.nih.gov/34617561/).

- The nutritional scheme adopted is not clear, a typical day should be clearly shown and how the plan was administered (generic indications or detailed program); nothing is said about the use of supplements

The following information is presented at page 3 lines 122-124: ‘During the seven weeks of the study, participants consumed a so-called low-impact diet, i.e. excluding gluten and having minimal consumption of FODMAPs, guided by a dietitian.’ Further details about the guidance by the dietitian is presented in a previous publication (https://pubmed.ncbi.nlm.nih.gov/34617561/)

The interventions foods are described in section 2.2 ’Dietary interventions’. Further details about the foods are presented in Supplementary Table 1 and 2. We have added that the interventions were served as powders, page 4, line 148: ‘in powder form’.

- Nothing is said about physical activity

Thank you, it is true that physical activity is not mentioned. There was no monitoring of physical activity during the trial since there are no clear indications that it is important for IBS symptoms. However, the day before visits to the clinic the participants were instructed to avoid vigorous physical activity, mostly to not perturb metabolism when blood sampling. The following sentence was added to the manuscript page 4, line 153-155: ’At each visit to the clinic the following was registered: adherence to overnight fasting routines and compliance to avoidance of vigorous physical activity and alcohol consumption during the preceding 24 h.’

- Possible food intolerances independent of IBS, which should be investigated before the intervention, should be considered

Thank you for the comment. Most participants showing interest in our study perceived themselves sensitive to FODMAPs/gluten or both, which reflects the situation for the majority of people with IBS. The double-blind elimination-exposure character of the study enabled an objective evaluation of tolerance to FODMAPs and gluten, hence, our results are of great value for health care personnel in the area.

Round 2

Reviewer 3 Report

The authors have greatly improved the manuscript. In my opinion, the center remains the microbiota with all the related limitations.

My note on the Bristol scale was to underline how, unfortunately, even today, although the methods used to evaluate the microbiota can be highly inaccurate, therefore basing the goodness of a method such as the FODMAP on something inaccurate is questionable.

It was improved, and probably a second revision is needed.

Author Response

Comments and Suggestions for Authors

The authors have greatly improved the manuscript. In my opinion, the center remains the microbiota with all the related limitations.

My note on the Bristol scale was to underline how, unfortunately, even today, although the methods used to evaluate the microbiota can be highly inaccurate, therefore basing the goodness of a method such as the FODMAP on something inaccurate is questionable.

Comments on the Quality of English Language

It was improved, and probably a second revision is needed.

Reviewer 3

Thank you for these important comments.

Indeed, there are several limitations with microbiota analysis. Such limitations are now more accurately discussed in the manuscript (page 16, lines 558-567):

‘Lastly, one must remember that there are several challenges with microbiota analysis. Until today the profile of a ‘normal’ microbiota is unknown, probably relating to context [99,100]. Importantly, there are several methodological challenges with micro-biota analysis and disparities of how to measure microbiota between studies [85]. In the present study, gut microbiota composition was assessed using 16S rRNA gene sequencing, whereas shotgun metagenomics could have contributed with more information on gut microbiota functionality and activity, which are known to differ between species and strains [29,32,97]. The challenges with microbiota analysis together with the heterogenicity of IBS [99], large placebo responses [87], and differences in study design [6] could relate to the large diversity reported in clinical studies.’

Your comment about Bristol scale, we agree, methods used to analyse microbiota have several limitations. However, we are now discussing these limitations in the manuscript and we consider it accurate to report our results, which are that FODMAPs increased microbiota related to beneficial health factors, which also are correlating to metabolites associated to beneficial health factors in previous studies. Therefore, we consider it accurate to report that ‘the minor effect of FODMAPs on IBS symptoms must be weighed against beneficial health effects’. We are also clearly stating that further clinical trials are needed.